# Melbournevirus encodes a shorter H2B-H2A doublet histone variant that forms structurally distinct nucleosome structures

Alejandro Villalta[1,2], Hugo Bisio [3], Chelsea M. Toner [1], Chantal Abergel [3] & Karolin Luger [1,2] ✉

Unique among viruses, some giant viruses utilize histones to organize their genomes into nucleosomes. Melbournevirus encodes a distinct H2B-H2A histone doublet variant in addition to the canonical H4-H3 and H2B-H2A doublets. This viral histone variant has a truncated H2B portion and its amino acid sequence deviates from that of the main viral H2B-H2A throughout the entire coding region. It is less abundant than the main H2B-H2A doublet, is likely essential for melbournevirus fitness, and is conserved in all *Marseilleviridae*. The cryo-EM structure of a nucleosome-like particle reconstituted with this H2B-H2A variant and viral H4-H3 reveals that only 90 base pairs of DNA are stably bound, significantly less than in eukaryotic nucleosomes and viral nucleosomes made with the main fused viral histone doublets. The reduced ability to bind DNA can be attributed to structural differences between variant and main H2B-H2A. Variant melbournevirus nucleosomes are less stable, possibly aiding rapid genome unpacking to initiate gene expression. Our results highlight the remarkable propensity of giant viruses to appropriate the utility of histones for their specialized purposes.

All eukaryotes organize their genomes into nucleosomes, a complex of four types of conserved histones (two copies each of H2A, H2B, H3, and H4) that wrap 147 base pairs of DNA[1]. Nucleosomes are the repeating structural unit of eukaryotic chromatin and contribute to the regulation of genome accessibility in major ways. This strategy, which was originally thought to be an exclusive feature of eukaryotes, is now increasingly found in non-eukaryotic organisms such as archaea, bacteria, and even viruses.

Most representatives of the domain of Archaea encode at least one minimalist histone that forms histone fold dimers with strong structural homology to the eukaryotic histone pairs H2A-H2B and H3-H4. Archaeal histones are much less conserved across the domain compared to their eukaryotic counterparts and typically lack N-terminal tails and other extensions[2,3]. Rather than forming defined particles, these histones organize DNA into slinky-like

hypernucleosomal structures of variable size[4–6]. Histone-like proteins were also identified in some bacterial clades, and in some cases, significant amounts of the histone are associated with the bacterial genome[7–9]. Recently, some double-stranded DNA viruses, particularly in members of the phylum *Nucleocytoviricota* (encompassing the "giant viruses"), were also found to encode histone-like proteins[10]. Nucleosome-like particles (NLPs) with histones encoded by the giant viruses melbournevirus and medusavirus have been structurally characterized[11–13]. While medusavirus encodes single copies of homologs for the four core histones H2A, H2B, H3, and H4, histones from representatives of the distantly related *Marseilleviridae* (including melbournevirus) are encoded as doublets, linking H4 to H3, and H2B to H2A (mel_368 and mel_369, respectively)[11–13]. The NLPs from both viruses exhibit distinct structural features and are more unstable in vitro than their eukaryotic counterparts. The essential nature of

[1]Department of Biochemistry, University of Colorado Boulder, Boulder, CO, USA. [2]Howard Hughes Medical Institute, University of Colorado Boulder, Boulder, CO, USA. [3]Aix–Marseille University, Centre National de la Recherche Scientifique, Information Génomique & Structurale, Unité Mixte de Recherche 7256 (Institut de Microbiologie de la Méditerranée, FR3479, IM2B), Marseille, France. ✉e-mail: karolin.luger@colorado.edu

histones has yet to be determined for histone-encoding viruses other than melbournevirus[12], and to date no evidence for epigenetic histone modifications has been found in the few giant viruses that have been studied.

In eukaryotes, the replacement of the canonical, replication-dependent core histones with histone variants (which are best described as "special histones for special occasions") contributes in significant ways to gene regulation and chromosome organization, by conveying unique features to the nucleosome[14]. Well-characterized histone variants include H2A.Z (associated with active transcription), H2A.B, and the H3 variant CenpA (located exclusively at centromeres of eukaryotic chromosomes)[14]. Deletion of the genes encoding histone variants is embryonic-lethal in most cases, indicating that the subtle structural differences of the variant nucleosomes convey specific functions. Histone variants have expression patterns that are distinct from the replication-dependent major-type histones, and they utilize separate mechanisms for their integration into nucleosomes[15].

Intriguingly, melbournevirus (MV) encodes a gene (mel_149; originally annotated as a "histone 2A-domain-containing protein") that we suspected to be the first virally encoded histone variant. The protein product was found in the virion of melbournevirus, together with the H4-H3 and H2B-H2A proteins, and was shown to be associated with Melbournevirus chromatin[12,16]. Proteomic analysis of the melbournevirus virion revealed that the mel_149 gene product is 30 times less abundant than H4-H3 (mel_368) and ~10 times less abundant than H2B-H2A (mel_369)[12]. Mel_149 is highly conserved across the entire *Marseilleviridae* family (Supplementary Fig. 1).

Here we show that mel_149 is likely essential for melbournevirus fitness. We determined the structure and thermal stability of an NLP reconstituted from H4-H3 (mel_368) and the putative H2B-H2A variant (mel_149). A structural comparison with our previously published structure of a nucleosome reconstituted with the two main melbournevirus histone doublets sheds light on the potential role of the H2B-H2A variant in the organization and regulation of the melbournevirus genome.

## Results

### Melbournevirus encodes a conserved shorter H2B-H2A doublet variant that is likely essential for viral fitness

We suspected that the mel_149 gene product might be a virally encoded H2B-H2A variant, as it has been shown to be associated with viral chromatin[16], and shares less than 31% identity with mel_369 (MV-H2B-H2A) and less than 23% identity with mel_368 (MV-H4-H3) (Supplementary Table 1). This gene has no recognizable homology to histone

H2B. Indeed, the H2A region of mel_149 is ~40% identical to H2A encoded by the amoeba host *Acanthamoeba castellanii* (*A. castellanii*), to *Xenopus laevis* H2A, and to the H2A region of MV-H2B-H2A (Supplementary Table 1). In contrast, the H2B portion does not have any similarity with *Xenopus laevis* H2B, *A. castellanii* H2B, or with the H2B region of MV-H2B-H2A. Additionally, no homolog for the isolated H2B portion was found in the entire *A. castellanii* genome. The Melbournevirus variant-H2B-H2A, with 168 amino acids (aa), is much shorter than the main MV-H2B-H2A doublet (269 aa). We will refer to the mel_149 gene product as MV-varH2B-H2A to distinguish it from MV-H2B-H2A.

Distinct features of the MV-varH2B-H2A amino acid sequence are its shorter predicted tail regions compared to MV-H2B-H2A (10 aa shorter H2B N-terminal tail, and 70 aa shorter H2A C-terminal tail), and the shortened H2BαC helix (Fig. 1). Additionally, the linker region between H2B and H2A in the variant is also predicted to be shorter by 10 aa. These features of MV-varH2B-H2A are highly conserved across the entire *Marseilleviridae* family (Supplementary Fig. 1), suggesting functional roles.

We previously demonstrated that the genes encoding MV-H4-H3 and MV-H2B-H2A (mel_368 and mel_369 gene products) are essential for Melbournevirus, suggesting key roles for these histone doublets in the viral life cycle[12]. To assess the importance of the putative variant H2B-H2A for viral fitness, we attempted to knock out mel_149 from the Melbournevirus genome, as described previously[12]. To enhance this approach, we introduced an N-acetyl transferase selection cassette, which has been shown to efficiently select recombinant mimivirus and mollivirus[17,18]. Despite effective inhibition of viral growth by nourseothricin, no variant H2B-H2A deleted viruses were recovered in two independent attempts, suggesting a strong phenotype associated with mel_149 deletion. To facilitate recombinant selection, we generated trans-complementing lines, as previously described[12]. This strategy enriched the variant H2B-H2A deleted virus population but did not yield clonal viruses in three independent attempts.

To further assess the impact of mel_149 deletion, we passaged the enriched variant H2B-H2A deleted virus fraction in the absence of selection, with and without trans-complementation. In all cases, variant H2B-H2A deleted viruses were rapidly lost, confirming the strong phenotype associated with mel_149 deletion. Importantly, this disappearance was not due to a simple bottleneck effect, as trans-complementing *A. castellanii* lines extended the number of passages of mixed progeny viruses containing knockouts (Fig. 2). Thus, the loss of variant H2B-H2A deleted viruses is likely due to the essentiality of mel_149 rather than due to a stochastic effect.

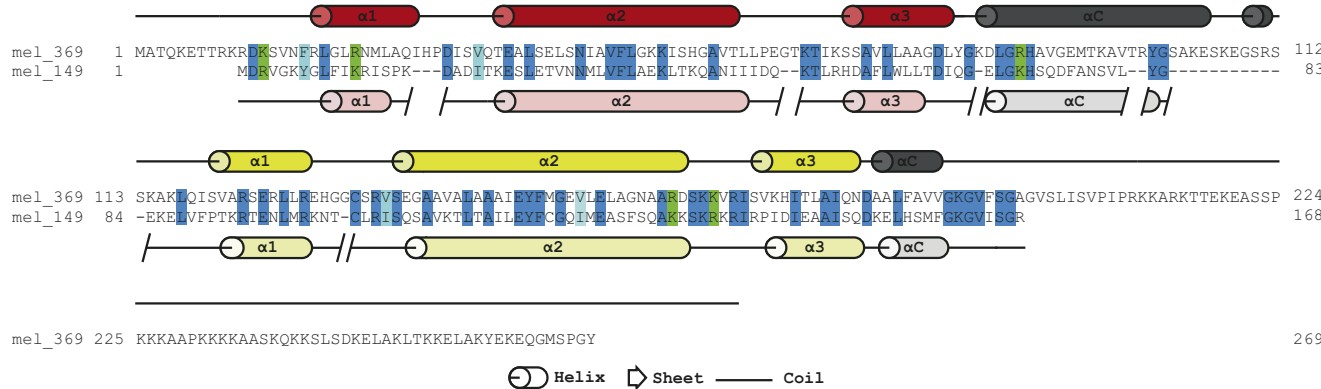

**Fig. 1 | Melbournevirus encodes an H2B-H2A histone variant.** The amino acid sequence of MV-H2B-H2A (mel_369, secondary structure elements taken from pdb 7N8N shown in dark red for H2B, and yellow for H2A) and MV-varH2B-H2A (mel_149, predicted secondary structure elements shown in light red and light yellow for H2B and H2A, respectively). Dark blue highlighted amino acids (aa) are conserved between the two proteins; similar hydrophobic amino acids are shown in light blue (V/I or F/W/Y), and conserved positively charged aa (R/K) are highlighted in green.

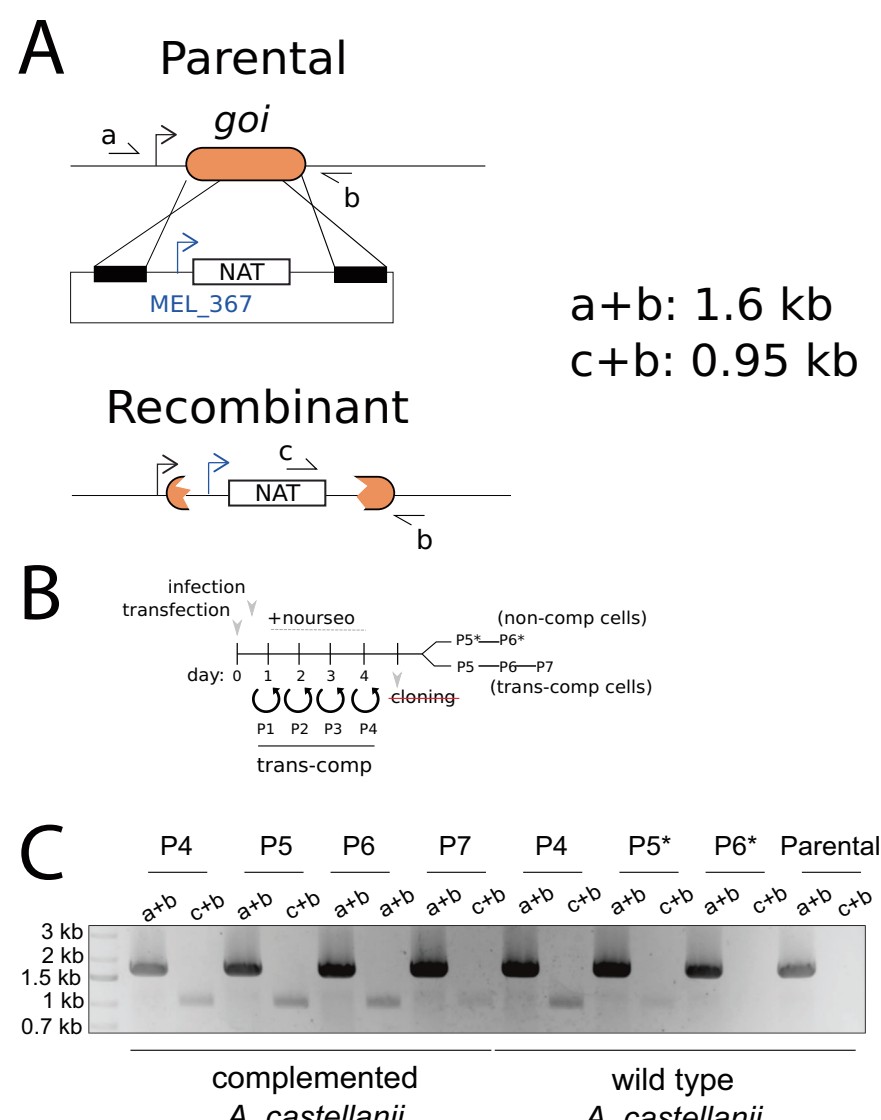

**Fig. 2 | mel_149 deletion strongly impacts melbournevirus fitness. A** Schematic representation of the strategy used to generate the variant H2B-H2A deleted virus. The selection cassette was introduced by homologous recombination. The location of the primers used for genotyping (a, b, and c) are indicated. **B** Variant H2B-H2A deleted viruses were generated, selected, and cloning was attempted as indicated by the timeline. After passage 4 in presence of selection (nourseothricin), since we were unable to obtain pure variant H2B-H2A deleted viruses; viruses were split and used to infect non-complemented amoebae (P5*) or trans-complementing amoebae (P5) in absence of selection. Trans-complementation refers to the restoration of the function of a deleted or defective gene in a viral genome through the expression of the same gene from an independent genetic element, such as the host genome or a plasmid. Subsequent passages were performed and genotyped to assess the fitness associated with gene knockout in competition with wild type viruses. **C** PCR amplifications of mel_149 at three cellular passages of *A. castellanii*, complemented with mel_149 or wildtype. Genotyping was performed using the primers indicated in (**A**) and the experiment was performed as shown in (**B**). This image is representative of 2 independent experiments.

## MV-varH2B-H2A and MV-H4-H3 assemble into structurally unique nucleosome-like particles (NLPs)

To investigate whether the putative histone variant MV-varH2B-H2A can form viral NLPs with MV-H4-H3, we used our standard procedure to assemble nucleosomes with a 147 bp DNA fragment (Widom 601 positioning sequence). MV-varH2B-H2A behaved like MV-H2B-H2A during reconstitution, and the two NLPs migrate similarly on a native gel (Supplementary Fig. 2A). Mass photometry analysis shows that the complex has a size of $157 \pm 33$ kDa, well within the error of the expected size of 133.5 kDa, confirming the assembly of the viral variant NLPs with a full complement of histones (Supplementary Fig. 2B).

Just as canonical MV-NLP, variant NLP is destabilized compared to eukaryotic nucleosomes, as demonstrated by a thermal shift assay[19]. This assay monitors the dissociation of histones from the DNA by measuring the exposure of hydrophobic amino acids to SYPRO Orange. MV-varNLP (with a melting temperature of $49.5 \pm 0.55\,°C$) is destabilized compared to MV-NLP ($52.8 \pm 0.4\,°C$), and this difference, although relatively modest, is statistically significant (Supplementary Fig. 2C–E). Both values are significantly lower than what is observed for eukaryotic nucleosomes, consistent with our experience that cross-linking is necessary to maintain intact viral NLPs on cryo-EM grids[12]. The human nucleosome was used as a proxy for a stably folded "canonical" nucleosome with a well-known structure. Both MV-NLPs are characterized by a denaturation curve that is qualitatively distinct from that of human nucleosomes. These typically show two distinct denaturing events: the first at around $75 \pm 0\,°C$ corresponding to H2B-H2A dissociation, and the second at $86.2 \pm 0.4\,°C$ corresponding to H4-H3 dissociation, as previously published[19]. In contrast, both variant and

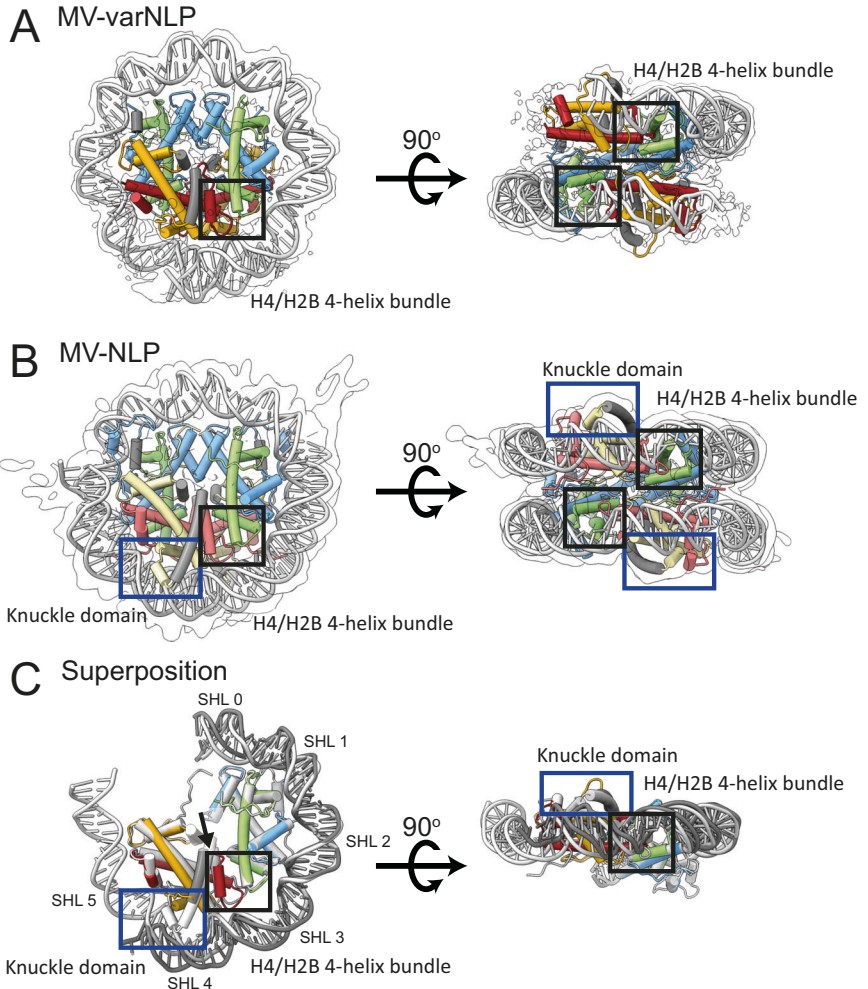

**Fig. 3 | MV-varNLP is structurally similar to MV-NLP.** Experimental electron density and model of (**A**) MV-varNLP, and (**B**) MV-NLP (PDB: 7N8N). The histone fold regions of H2B are shown in red, H2A in yellow, H4 in green, and H3 in blue, with helices not belonging to the histone fold in grey. **C** Superposition of MV-varNLP with MV-NLP. Here, only half of the nucleosome is shown for clarity. Superhelix locations (SHL) are numbered as previously defined[1]. The four-helix bundle region tethering H4 to H2B, and the "knuckle domain" that binds DNA in MV-NLP are indicated by black and blue boxes, respectively. The arrow corresponds to the H2B α3-αC helix interface.

canonical MV-NLPs dissociate either in a single step or in two concurrent steps that are indistinguishable in our assays (Supplementary Fig. 2C, D).

To stabilize variant-NLP for cryo-EM, we crosslinked samples using a GraFix protocol that we previously optimized for MV-NLP[12]. We were able to determine the structure of MV-varNLP to an overall resolution of 4.4 Å (Supplementary Fig. 3 and Supplementary Table 2), with higher resolution observed for the protein core, while parts of the DNA and the tail regions are less well defined. MV-varNLP contains the requisite two copies of MV-H4-H3 and MV-varH2B-H2A (although one copy is better resolved than the other), and binds DNA in the canonical superhelical configuration typical for all nucleosomes. However, only about 90 bp of DNA are represented by density in MV-varNLP, compared to 124 bp in the MV-NLP density, and 147 bp in most canonical eukaryotic nucleosome structures (Fig. 3). This is consistent with previous observations that MV-varNLP are more sensitive to MNase digestion[16].

To exclude that the observed structure is an artifact of the strong nucleosome positioning sequence (Widom 601) used here, we also assembled MV-varNLP on a 150 bp "random-sequence" DNA (GC-content of 50% which is close to the 45% GC-content of Melbournevirus genome). Nucleosomes are similar to what we observed for 601 sequence (Supplementary Fig. 4A). We collected a small dataset on crosslinked particles. The 2-D class averages look remarkably similar to

the particles formed with 601 DNA (Supplementary Fig. 4B–D). However, the heterogeneity of histone positioning on DNA sequences other than strong positioning sequences (a general phenomenon which led to the prevalence of 601 sequence in published nucleosome structures) made particle alignment difficult, precluding structure determination. We therefore proceeded with MV-varNLP made with 601 DNA.

In our structure, density for the first 14 amino acids are missing for the MV-H4-H3 portion, and the entire 30 amino acid connector between H4 and H3 is too disordered to confidently be built from the map, even though these regions were visible in our previously published maps of MV-NLP[12]. MV-varH2B-H2A is well resolved, and only the first four amino acids are not represented by density. Overall, the histone portions of variant and major-type MV-NLPs superimpose with a root mean square deviation of the overall structure (r.m.s.d.) of 1.76 Å. This can be mostly attributed to structural differences between the H2B-H2A portion (r.m.s.d. of 1.6 Å), while the H4-H3 portion, where the amino acid sequence is identical between the two structures, exhibits closer similarity (r.m.s.d. of 1.2 Å; Fig. 3C).

The incomplete wrapping of DNA in MV-varNLP is caused by sequence differences between variant and main H2B-H2A. The H2B region of the variant histone doublet provides the requisite configuration for a stable four-helix bundle interface with the H4 portion of

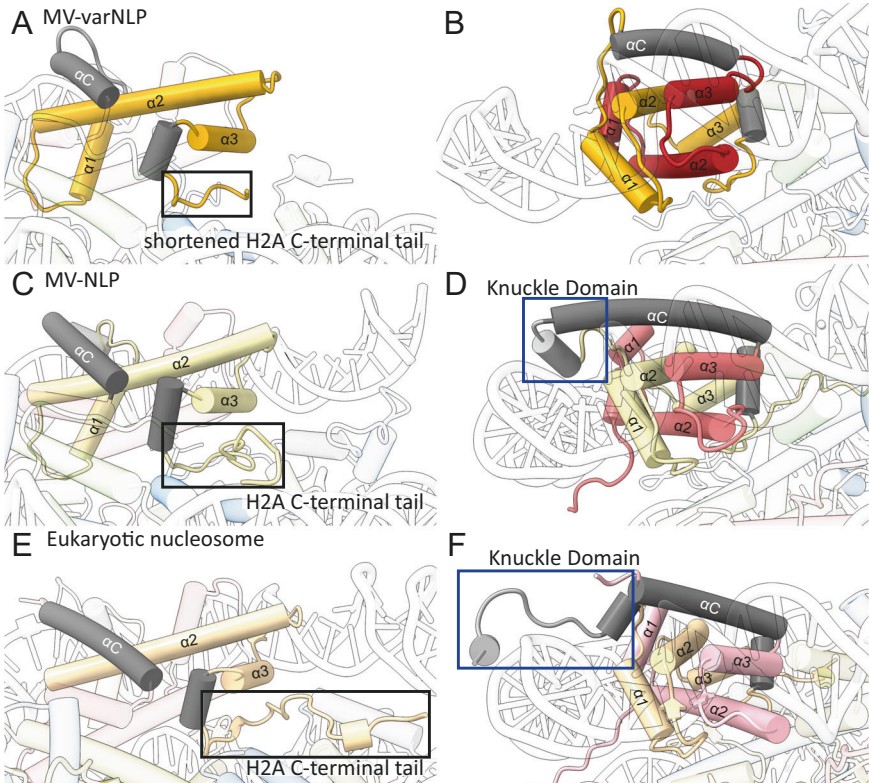

**Fig. 4 | MV-varNLP and MV-NLP differ in their H2B portion and in the extent of DNA binding.** Cartoon representation of parts of H2B and the H2A C-terminal tail and knuckle domain of (**A**, **B**) MV-varNLP, (**C**, **D**) MV-NLP (7N8N), and (**E**, **F**) the eukaryotic nucleosome (1AOI). Models are represented in transparent colors with solid colors at regions of interest (H2A, yellow; H2B, red; knuckle domain, gray).

MV-H4-H3, and strong contacts with the DNA are established at this site (SHL ± 3.5, Fig. 3C). The L2 loop connecting α1 to α2 in MV-varH2B is shorter by two amino acids but maintains interactions with its MV-varH2A L1 counterpart and with DNA. This is the last contact observed between histones and DNA in MV-varNLP (Fig. 3C). The MV-varH2B α3 helix is one turn shorter than in other H2B-like structures, and the adjacent αC helix, which defines the surface of the nucleosome, is two turns shorter than in any other H2B known to date. The interface between it and the H2B α3 helix is less pronounced, and it stops short of extending to the DNA (Fig. 3C, arrow). The orientation of MV-varH2B α1 has shifted. Its location is stabilized by a hydrophobic core composed of amino acids from the connector linking H2B with H2A that is distinct from MV-H2B-H2A and particularly conserved among members of the *Marseilleviridae* variant histones (Supplementary Fig. 1). This helix is also shorter by one turn (Figs. 3 and 4), due to a highly conserved Proline (P16) that terminates it prematurely, and the loss of contacts with the H2A α2 helix in the histone fold arrangement. This causes the MV-varH2B α2 helix to run straight rather than bent as it is in most other nucleosome structures (Fig. 4A, B).

The connector between H2B and H2A is much shorter in the variant histone and fails to form the "knuckle domain" that in MV-H2B-H2A contact DNA at SHL 4.5. This domain, which is also present in H2A of all eukaryotic nucleosome structure including those with H2A variants[20,21], is formed by a short additional α helix that points its N-terminal end at the DNA (Fig. 4). Due to its absence in MV-varH2B-H2A, no contacts with DNA are established at SHL 4.5, even though this region maintains an overall positively charged surface (Supplementary Fig. 5).

The H2A α1 helix in MV-varH2B-H2A exhibits a different conformation compared to MV-H2B-H2A. This region differs substantially in its amino acid sequence from the major-type histone, and is particularly well conserved among members of the *Marseilleviridae*

(Supplementary Fig. 1). The helix bends approximately 17° towards the DNA, positioning its N-terminus to contribute to DNA binding through its positive dipole moment (Fig. 4). This might help compensate for the lack of interaction that would otherwise be provided by the longer H2B αC helix and the knuckle domain in canonical H2B-H2A, thereby maintaining some level of DNA binding and stabilization in the variant nucleosome. Indeed, weak density for the conventional DNA superhelical path is observed extending to SHL 5.5, suggesting residual DNA interactions (Supplementary Fig. 5E, F).

Interestingly, a conserved cysteine in the MV-varH2A L1 loop forms a disulfide bond with the second varH2A, supported by well-defined electron density (Supplementary Fig. 6A), seemingly stapling together the two MV-varH2B-H2A variant moieties. This cysteine is also present in main MV-H2A but due to differences in loop configuration, does not engage in disulfide bond formation, as shown in two independently determined structures[11,12]. We were able to observe weak evidence for this disulfide bridge in solution within the MV-variant NLP (Supplementary Fig. 6B), but the functional significance of this stapling interaction is currently unknown. This feature is not seen in any eukaryotic H2A histone sequence. Finally, the docking domain in MV-varH2A maintains its characteristic orientation in the portion that packs against the C-terminal region of H4 to stabilize the interaction with H4-H3. However, the "stem" portion in MV-H2A and eukaryotic nucleosomes that provides further contacts with H4-H3 is missing in all viral variant H2A sequences, leading to a less-stable interaction of MV-varH2B-H2A with MV-H4-H3, and an altered nucleosome surface (Figs. 3C and 4 and Supplementary Fig. 5).

**Melbournevirus NLPs can stack to form stable di-nucleosomes in silico**

It has been suggested that melbournevirus nucleosomes are tightly packed within the virus with very little connecting linker DNA, possibly

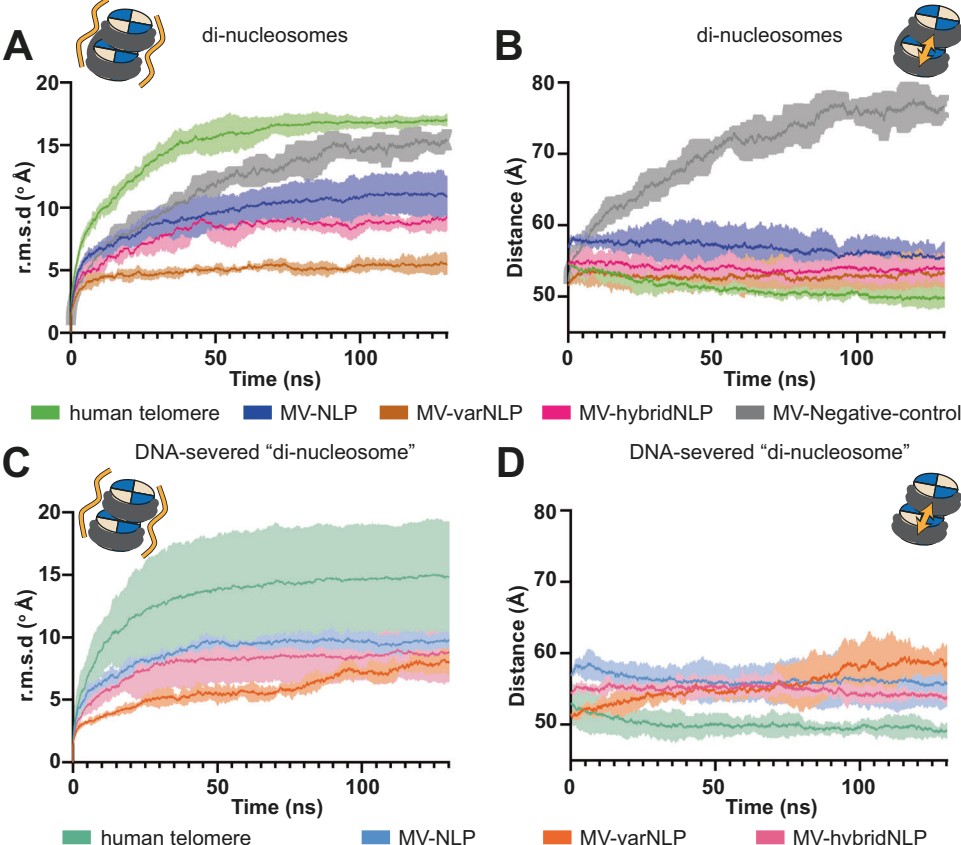

**Fig. 5 | MV-NLP can form nucleosome stacks in silico.** Two parameters were measured to assess the stability of di-nucleosome stacks: The mean variation of (**A**, **C**) the root mean square deviation (r.m.s.d) of the overall structure compared to the start of the simulation, and (**B**, **D**) distances between the center of gravity of the two nucleosomes over the simulation time are plotted for the human telomeric di-nucleosome (green), MV-NLP (blue), MV-varNLP (brown), MV-hybridNLP (purple) and a control for MV-varNLP wherein inter-nucleosomal protein-protein contacts were mutated in silico to Glu (grey). **C**, **D** show a simulation of di-nucleosomes

where 10 bp at the DNA linker portion connecting both nucleosomes were removed to sever the connection between the two particles. The larger r.m.s.d variations at the beginning of the simulation for the human telomeric di-nucleosome are due to the long tails that collapse onto the DNA, as previously observed for eukaryotic mono-nucleosomes[21]. The lighter area around the mean curve corresponds to the standard variation around the mean curve, from three replicates. Individual tracks are found in Supplementary Fig. 7.

forming a structure similar to the telomeric structure in eukaryotes[16]. Human telomeres are characterized by exceedingly short nucleosome repeat length, forcing a "columnar stacking" of nucleosomes rather than the more canonical "beads on a string" arrangement observed in genomic nucleosomes[22]. This organization could provide further stabilization in the virions. Therefore, we decided to conduct molecular dynamics simulations, to check whether this stacking is compatible with both melbournevirus nucleosomes structures. Human telomeric di-nucleosomes (PDB: 7V96) represent a good tentative starting model for this type of arrangement. Because tails were absent in that structure, we replaced the histones in the telomeric di-nucleosome structure (7v96) with a model of histones predicted with Alphafold, and manually adjusted the tails to avoid direct clashes, to have a complete model without the missing regions of the experimental models, These initial models were largely similar to the experimentally determined structure (r.m.s.d of 1.84, 1.06, and 1.27, for MV-NLP, MV-varNLP and human histone octamer, respectively). We conducted all-atom molecular dynamics simulations to assess the in silico stability of the resulting viral di-nucleosomal stacks. In this configuration, both MV-NLP and MV-varNLP are stable over time and remain stacked (Fig. 5A, B). This is despite differences in the makeup of the nucleosomal surface, caused by the divergent H2B αC helix, which in the model is positioned to stabilize the stack. The H2A lysine-rich C-terminal tail, which in the predicted MV-NLP di-nucleosome model also contributes to stacking stability, is missing in MV-varNLP. As such, the MV-varNLP

di-nucleosome stack is more affected in the simulation by severing the DNA than MV-NLP or telomeric eukaryotic nucleosomes, suggesting weakened histone-histone stacking interactions (Fig. 5C, D). Deleting the histone H4 tail of MV-H4-H3 impacts the stability of the viral di-nucleosome stacks, according to the increase in distance between the two nucleosomes in the course of the simulation (Supplementary Fig. 7F and L). Although this effect is somewhat stochastic, it is on average larger than severing the DNA connecting the two nucleosomes (compared to Supplementary Fig. 7D and J). We also performed a simulation of a hybrid MV-varNLP•MV-NLP di-nucleosome stack. This hybrid di-nucleosome was also stable over time, and its behavior lies between that of MV-NLP and MV-varNLP (Fig. 5D and Supplementary Fig. 7S–V). Importantly, a control in which we mutated (in silico) all contacts between nucleosomes to Glu is unable to maintain di-nucleosome stacking during simulation (Fig. 5B, grey curve), demonstrating that this experiment can indeed distinguish between compatible and incompatible nucleosome stacking surfaces. Together, this data suggests that despite differences in surface charge and contours, both MV-NLP and MV-varNLP can form closely packed nucleosomal stacks. As with all simulations and predictions, their existence and precise molecular composition must be verified experimentally.

## Discussion
Since their discovery, "giant viruses" have continued to surprise the scientific community with their remarkable size, complex life cycles,

and unique genetic features that are characterized by a large number of ORFans (genes with no known homologs or function). Even viral proteins with assigned functions exhibit unexpected and intriguing properties. A prime example is the presence of genes encoding histones, histone variants, and in some instances even putative linker histones in a subclass of the *Nucleocytoviricota*[13,23]. The fusion of histones into doublets in Melbournevirus likely facilitates the obligatory pairing of histones H2B with H2A, and histones H4 with H3, perhaps to decrease the reliance on histone chaperones. Recent analyses of many giant virus metagenomes unearthed a wide variety of histone fusions, indicating that this might not be an isolated incident[10].

Here we show that melbournevirus encodes an H2B-H2A variant in addition to the highly abundant main H2B-H2A doublet. Both H2B-H2A doublets, as well as the H4-H3 doublet, are likely essential for viral fitness, as deletion of any of the three genes results in virus that is not infectious (this study and ref. 13). This indicates that the variant H2B-H2A histone doublet has functions that cannot be fulfilled by the main histone H2B-H2A doublet or by *A. castellanii* host histones, despite its much lower abundance in the virus. Genes for this variant are present across the entire *Marseilleviridae* family and its distinct features are highly conserved. The nucleosomes that it forms with the H4-H3 doublet are structurally distinct from the canonical melbournevirus NLPs. The histone H2B-H2A variant has an H2B region that can be barely classified as such based on its sequence, yet assembles into a protein core with the full complement of histones, with the characteristic positively charged helical ramp around its outside. Nevertheless, this protein core stably binds only ~90 bp of DNA, and the terminal 25 base pairs of DNA are not represented by density in our maps. The diminished contacts between DNA and protein beyond SHL ± 3.5 are due to a shorter H2B αC helix, a redirected and shorter connector between H2B and H2A, and by a missing portion of the H2A docking domain that in other structures stabilizes the H3 αN helix to interact with the penultimate 10 bp of DNA. Lower degrees of wrapping of in vitro assembled variant MV-NLPs were also observed by others through MNase protection assays[16]. We would have expected these features to decrease the overall stability of MV-varNLP (compared to MV-NLP) to a higher degree than what we determined experimentally. We hypothesize that the presence of the conserved disulfide bond, stapling together the two MV-varH2B-H2A moieties, and not present in MV-H2B-H2A or any other H2A known to date, might counteract these potentially destabilizing features.

In eukaryotes, histone H2A variants in particular are known for forming nucleosomes that organize less DNA (see, for example, the human histone H2A variant H2A.B; PDB: 6M4H[20]), and as such, the melbournevirus H2B-H2A variant fits into this category of nucleosome-destabilizing histone variants. Intriguingly, when determining the structure of nucleosomes containing the human histone variant H2A.B, an artificial H2B-H2A.B doublet had to be made, and the resulting nucleosomes had to be further stabilized by GraFix prior to structure determination by cryo-EM[20]. As such, the fusion of histones into doublets, observed as a natural occurrence for all histones in melbournevirus, might aid nucleosome assembly in absence of histone chaperones and assembly factors. This is consistent with the finding that multi-histones organization of metagenomic giant virus histone sequences appears to facilitate self-assembly of nucleosomes in *E. coli*[10].

In a biological context, nucleosomes are surrounded by other nucleosomes and form closely packed higher order structures. In eukaryotic chromatin, genomic nucleosomes are separated by linker DNA ranging in length between 15 and 65 bp[24], resulting in a beads-on-a-string appearance of nucleosomal arrays in their most relaxed states. The Melbournevirus genome appears to be densely packaged by nucleosomes with an exceedingly short repeat length of 121 bp, without connecting linkers and apparently not even providing the requisite 147 bp for a full nucleosomal superhelix[16]. This could result in stacking of nucleosomes to promote DNA-histone contacts beyond the 90 bp observed in the structure, as the DNA would already be partially constrained and twisted by neighboring nucleosomes. This could compensate for the lack of protein-DNA interactions beyond SHL ± 4.5. Indeed, our MD simulations suggest that both canonical and variant Melbournevirus nucleosomes can stably stack as di-nucleosomes similar to human telomeric nucleosomes, despite pronounced differences in nucleosomal surface and histone tails[16]. As such, it appears that the histone surfaces of both variant NLPs maintain the ability to form closely packed nucleosomal arrays. It is however, important to point out that the conclusions from these simulations must be verified experimentally.

In the melbournevirus capsid, MV-varH2B-H2A (mel_149) is ~10 times less abundant than the main H2B-H2A doublet (mel_369). This means that MV-H2B-H2A is the main constituent of viral chromatin, while MV-varH2B-H2A is either randomly interspersed or clustered in the genome. Unfortunately, experimental maps of the genomic distribution of MV-varH2B-H2A are not yet available. Difficulties in obtaining ChIP-quality antibodies that are specific to viral histones, and the inability of generating viable viruses with tagged histone doublets have hampered these efforts.

Members of the family *Marseilleviridae* possess all genes necessary to transcribe their genes and cap their mRNA, but do not carry the RNA polymerase subunits in their capsids. Instead, during the early infectious cycle, they destabilize the host nucleus and transiently recruit the host machinery to the early viral factory until the virally encoded RNA polymerase is translated and localized to the viral factory[25]. We hypothesize that MV-varH2B-H2A forms nucleosomes over genes that need to be expressed early in the viral cycle, and that the rapid dissociation of MV-varNLP upon host infection allows speedy transcription by the host polymerase of early genes. Such a role in orchestrating gene transcription would be in keeping with the function of many eukaryotic histone variants.

Proteins organizing DNA in recently studied giant viruses seem to share a common feature of inherent instability. For instance, the mimivirus genomic fiber is designed to disassemble quickly to unlock the RNA polymerase and initiate rapid gene transcription, facilitating the viral propagation cycle[26]. Medusavirus encodes histones to organize its DNA, and like Melbournevirus, its nucleosomes are less stable than eukaryotic nucleosomes[13]. We hypothesize that DNA organizing proteins in giant viruses have adapted to package and protect the genome within the capsid, and to rapidly dissociate once the viral infection cycle has begun.

To date, it is unknown whether any of the viral histones harbor post-translational modifications that in eukaryotes are used for gene expression regulation, even though it is noteworthy that Melbournevirus does encode for a methyltransferase. It is also unknown whether host-encoded ATP-dependent chromatin remodelers are recruited from the compromised host nucleus and are utilized to free up the genome, or whether viral nucleosomes, unlike their eukaryotic counterparts, are self-assembling and self-organizing. Whether the H2B-H2A is placed randomly or whether it is targeted to specific regions of the viral genome by dedicated assembly factors remains an open question. It is entirely possible that novel components of a viral chromatin maintenance machinery are hiding amongst the many viral ORFan genes whose functions are currently unknown.

## Methods

### Generation of vectors

**Melbournvirus mel_149 knockout vector.** Gene knockout strategy was performed as previously described for pandoravirus and mimivirus[17,27]. The plasmid for gene knock-out was generated by sequential cloning of the 3′ UTR of mel_282, the promoter of mel_367, and a neomycin (NEO) selection cassette. Each cloning step was performed using the Phusion Taq polymerase (ThermoFisher) and

InFusion (Takara). Finally, 500-bp homology arms were introduced at the 5′ and 3′ end of the cassette to induce homologous recombination with the viral DNA[18]. Before transfection, plasmids were digested with EcoRV and NotI. All primers are shown in (Supplementary Table 3).

**MV-var-H2B-H2A expression vectors.** the mel_149-GFP plasmid previously described[12] was modified as follows: (1) the neomycin selection cassette was replaced for a nourseothricin N-acetyl transferase (NAT) using the primers listed in Supplementary Table 3. (2) A stop codon was added between the mel_149 and the GFP to allow expression of mini H2B-H2A without a tag using the primers in Supplementary Table 3.

### Cell transfection and generation of trans-complementing lines
$1.5 \times 10^5$ A. castellanii cells were transfected with 6 µg of MV-var-H2B-H2A encoding plasmid using Polyfect (QIAGEN) in phosphate saline buffer (PBS) according to the manufacturer's instructions. Selection of transformed cells was initially performed at 30 µg/mL Nourseothricin and increased up to 100 µg/mL within a couple of weeks. The procedure is described step by step in ref. [18].

### Gene knock-out
Gene knockout was performed following our published protocol[18]. Briefly, $1.5 \times 10^5$ A. castellanii cells were transfected with 6 µg of linearized plasmid using Polyfect (QIAGEN) in phosphate saline buffer (PBS). One hour after transfection, PBS was replaced with PPYG, and cells were infected with $1.5 \times 10^7$ melbournevirus particles for 30 min with sequential washes to remove extracellular virions. 24 h after infection, the new generation of viruses (P0) was collected and used to infect new cells. An aliquot of P0 viruses was utilized for genotyping to confirm the integration of the selection cassette. Primers used for genotyping are shown in Supplementary Table 3. The new infection was allowed to proceed for 30 min, then cells were washed to remove extracellular virions and geneticin was added to the media. Viral growth was allowed to proceed for 24 h. This procedure was repeated one more time before removing the geneticin selection to allow viruses to expand more rapidly. Once the viral infection was visible, the selection procedure was repeated one more time. Viruses produced after this new round of selection were used for genotyping and cloning, as previously described[18].

### Melbournevirus histones doublet purification
Mel_149 and mel_368 were cloned into a pET-28 plasmid to add an N-terminal 6-His tag. Vectors were transformed onto E. coli (DE3) Rosetta cells for expression after induction with 0.4 mM IPTG at 37 °C, for 2.5 h. Pellets from 1 L of E. coli cells were re-suspended and lysed with 40 mL of guanidinium lysis buffer (6 M Guanidinium HCl, 40 mM HEPES pH 6.8 and 500 mM NaCl) and sonicated four cycles for 30 s at 50% strength (Branson Digital Sonifier). Lysate was spun at 16,000 rpm for 25 min using a JA-20 rotor (Beckman) on an Avanti J-20 XPI (Beckman). The supernatant was incubated with Nickel NTA Agarose Beads (GoldBio) for ~ 30 min at room temperature (RT). Nickel beads were pelleted at 1500 rpm for 2 min, the supernatant was discarded, and beads were resuspended with denaturing binding buffer (8 M Urea, 40 mM HEPES, 200 mM NaCl, pH 6.8). After gentle shaking for 2 h at RT, beads were washed with denaturing wash buffer (8 M Urea, 40 mM HEPES, 200 mM NaCl, 200 mM Imidazole, pH 6.8). Proteins were batch-eluted using histone elution buffer (8 M Urea, 20 mM HEPES pH 6.8, 200 mM NaCl, and 500 mM imidazole). To refold histones doublets, samples were dialyzed against 1 M urea Buffer (1 M urea, 20 mM HEPES pH 6.8, 200 mM NaCl, pH 6.8) for 6 h, followed by overnight dialysis against the same buffer without urea (1 mM EDTA, 20 mM HEPES, and 200 mM NaCl, pH 6.8). Precipitated protein was removed by centrifugation, and the supernatant was concentrated and run over a size-exclusion column S200 16/60 (Cytiva) equilibrated in 20 mM HEPES and 200 mM NaCl, pH 6.8. Samples were stored in 20% glycerol at −80 °C.

### Nucleosome reconstitution
Purified and refolded MV-varH2B-H2A (or MV-H2B-H2A as controls), MV-H4-H3 and a 147 bp DNA fragment (601 Widom) were mixed at a ratio of histone doublet to DNA = 2:2:1. Samples were dialyzed overnight against a gradient buffer (2–0.25 M NaCl, 10 mM MES, 1 mM DTT, pH 6.5) starting at 2 M NaCl and going to 0.25 M NaCl with an exchange flowrate of 1.5 ml/min using an Econo gradient pump (BioRad). The reconstituted products were analyzed by 5% native-PAGE.

Nucleosomes were also reconstituted with a "random 150 bp DNA" that had been designed to exhibit 50%GC-content over each 10% segment. In this and subsequent steps, both nucleosomes were treated identically, irrespective of the DNA sequence.

DNA sequence:

5′-GCTAGTCCGTCTTCTACTCTGAAATGAGCAGTCCTAGTCAG-CAAGATCG CTCAGCCAACTTTCTACCAGCGCAACCCTAATCTACCC-CATGAATGAAGCCGCACCCAAAACCGCATTCTAAGGAGTGA-CATTAACCCTCGGTGAGGATGT-3′.

### Sucrose gradient and crosslinking (GraFix)
A linear 5–40% sucrose gradient was prepared by pouring 6 ml of 5% sucrose buffer (50 mM NaCl, 20 mM HEPES, 0.1 mM EDTA, pH 6.5, 5% sucrose, and 0.16% glutaraldehyde) into a centrifugal tube (Beckman, 331372). 6 mL 40% sucrose buffer (50 mM NaCl, 20 mM HEPES, 0.1 mM EDTA, pH 6.5, and 0.16% glutaraldehyde) was slowly added from the bottom. The tubes were spun on a gradient maker (BioComp Gradient Master) to form a continuous gradient. Then, 200 µl of sample was loaded on top and spun at 4 °C for 18 h at 30,000 rpm (Beckman, Rotor SW-41Ti). The gradient was fractionated using a FC 203B Fraction collector (Gilson). The fractions were assessed by 5% native-PAGE. The nucleosome-containing fractions were then pooled and dialyzed against 50 mM NaCl, 20 mM MES, 1 mM EDTA, pH 6.5, and 1 mM DTT to remove the sucrose.

### Cryo-electron microscopy (cryo-EM) single particle data collection and analysis
Crosslinked nucleosomes were concentrated to 2–3 µM using Amicon Ultra-4 centrifugal filters (Ultracel 50 K, Millipore). 4 µl of the concentrated sample were applied to glow discharged (40 mA, 30 s, EMItec, Lohmar, DE) C-Flat 1.2/1.3 (Cu) grids, and blotted for 4 s using a Vitrobot Mk IV (Thermo Scientific) using the following parameters: 4 °C, 100% humidity, blotting force 1, wait time 30 s, and plunge frozen in liquid ethane cooled to liquid nitrogen temperature. Images were collected at nominal magnification of $130,000 \times g$ (Pixel size: 0.97 Å) on a FEI Titan Krios (300 kV), equipped with a Falcon 4 Direct Detection Camera. The movies were captured in super resolution mode with a total dose of 50 e/Å² (electrons per Angstrom square). Defocus range was −1.0 to −2.5 µm. The dataset was pre-processed (motion correction and CTF estimation) and processed with CryoSPARC (v4)[28,29]. Particles were picked using Blob picker. Then 1,010,995 particles across 2894 micrographs were extracted with a 448-pixel size box (up-sampling factor 2) after Inspect Picks. Particles were 2D classified into 400 classes, where only 12 classes were kept (85,405 particles). Five ab initio models were generated, followed by heterogeneous refinement, non-uniform refinement, as well as local refinement of the best ab initio model (36,051 particles), were performed to improve the density resolution to 4.41 Å, as estimated by GFSC.

### Model building and refinement
Initial protein models were built using AlphaFold 2, which were fitted into the final 3D electron maps in UCSF ChimeraX[30]. These models were then iteratively refined by multiple runs in COOT[31] and PHENIX[32].

For the DNA, DNA from 7N8N was used as a starting model, followed by further iterative refinement in PHENIX.

## Molecular dynamics

Initial models used for molecular dynamics simulations were made by taking the experimentally solved DNA model (7v96) with a histone predicted models on Alphafold 2, to have a complete model without the missing regions of the experimental models[33]. The terminal tails were manually adjusted in ChimeraX[30] to avoid clashes with the DNA. These initial models were compared to the experimentally determined and were largely similar (r.m.s.d of 1.84, 1.06, and 1.27). Then, all-atom molecular dynamics simulations using explicit solvent were carried out using AMBER18[34] using the ff14SB, bsc1, and tip3p forcefields (for protein, DNA, and water, respectively). Structures were protonated and hydrogen mass repartitioned (as implemented through "parmed" in AMBER). Structures were placed in cubic boxes surrounding the structures by at least 25 Å, charge neutralized using potassium ions and hydrated with water molecules. The structures were energy minimized in two 5000-step cycles, the first restraining the protein and DNA molecules to allow the solvent to relax and the second to allow the whole system to relax. Minimized structures were then heated to 300 K and slowly brought to 1.01325 atm. These systems were then simulated for 130 ns in 4 femtosecond steps. Simulations were carried out on NVIDIA GPUs (RTX6000s or A100s) using CU Boulder's Blanca Condo cluster. Root mean square, deviations (RMSD), distance, angle, and dihedral analysis were carried out using cpptraj through AMBER18[34]. Each simulation was carried out in triplicate; individual trajectories are shown in Supplementary Fig. 7. As a negative-control, we prepared (in silico) a MV-varNLP where we mutated the residues involved in contacts between the two nucleosomes of the di-nucleosome to Glu (H4-H3: 1–23; 94–131, and 157–160; varH2B-H2A: 66–72).

## Structural characterization of the variant-H2B-H2A/H4-H3 nucleosome model

Comparisons between nucleosome structures were conducted using UCSF ChimeraX[30]. The r.m.s.d between different models was calculated using COOT[31]. Marseilleviridae histones sequence alignments were Marseilleviridae histones sequence alignments were performed using MAFFT software[35].

## Thermal shift assay

For thermal shift assays, we mixed nucleosome preparations at 2.25 μM with 5× SYPRO Orange in 20 μl final reaction mix, in 20 mM MES, 50 mM NaCl at pH 6.5. We used StepOnePlus Real-Time PCR unit (Applied Biosystems) to increase the temperature from 25 °C to 95 °C at a rate of 1 °C/min and measured the fluorescence at 560 nm every 1 °C. Normalized data was plotted by displaying relative fluorescence over temperature using GraphPad Prism version 10.3.1 for Windows (GraphPad Software, Boston, Massachusetts, USA). The thermal melting point (Tm) ranges of each sample and replicate measured were determined by identifying the temperature at the lowest point of the fluorescence derivative (−d/dU RFU). Statistical analysis (independent 2-sample $t$-test with equal variance) was performed with GraphPad Prism version 10.3.1 for Windows.

## Reporting summary

Further information on research design is available in the Nature Portfolio Reporting Summary linked to this article.

## Data availability

The pdb coordinates and EM maps have been deposited in the PDB under accession code 9CVT; EMD-45966. The MD simulation data (Movies, initial models, distance and rmsd values) are available at figshare, with a link provided in the Supplementary Information/Source Data file. The data for the MD trajectories are available on request, they have not been uploaded due to the large size of files. Vectors are available on request. Source data are provided with this paper.

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

## Acknowledgements

We thank Garry Morgan and Johannes Rudolph for help with data collection, and Shawn Laursen for help with data analysis. Data collection was done at the Biochemistry Krios Electron Microscopy Facility (Bio-KEM) at CU Boulder (RRID: SCR_019057). We thank the Shared Instruments Pool (SIP) core facility (RRID: SCR_018986), Department of Biochemistry, University of Colorado Boulder, and Dr. Annette Erbse for the use of the shared research instrumentation infrastructure. The software used in this project was curated by SBGrid. This work utilized the CUmulus on-premise cloud service at the University of Colorado Boulder. CUmulus is jointly funded by the National Science Foundation (award OAC-1925766) and the University of Colorado Boulder. This work utilized the Blanca condo computing resource at the University of Colorado Boulder. Blanca is jointly funded by computing users and the University of Colorado Boulder. Data storage is supported by the University of Colorado Boulder PetaLibrary. This study was funded by the Howard Hughes Medical Institute (A.V., C.M.T., and K.L.) and the European Research Council (ERC) under the European Union's Horizon 2020 research and innovation program (grant agreement No 832601; H.B. and C.A.). The content is solely the responsibility of the authors and does not necessarily represent the official views of the funding agencies.

## Author contributions

A.V.: research conceptualization and design; research; data curation; validation; analysis; writing (original draft, review, and editing); visualization. H.B.: research conceptualization and design; research; data curation; validation; analysis; visualization. C.M.T.: research conceptualization; data curation; writing (review, and editing). C.A.: research conceptualization; funding acquisition; writing (review, and editing). K.L.: research conceptualization and design; funding acquisition; project administration; validation; analysis; writing (original draft, review, and editing), visualization.

## Competing interests

The authors declare no competing interests.
