## [Transparent Peer Review file · Nature Communications]

Melbournevirus encodes a shorter H2B-H2A doublet histone variant that forms structurally distinct nucleosome structures

Corresponding Author: Professor Karolin Luger

Version 0:

Reviewer comments:

Reviewer #1

(Remarks to the Author)

The nucleosome is the basic structural unit of the eucaryotic chromosomes. The author's group has been pioneering the structural biology of the nucleosomes and nucleosome-containing complexes. The authors and other groups have recently found that non-eucaryotes, such as Archaea and Marseillevirus, also have nucleosomes (PMID: 28798133, 34297924, 33927388). In the Marseillevirus, histones are encoded as doublets, linking H4 to H3 and H2B to H2A, and form nucleosomes that only wrap around ~120 bp DNA. In this manuscript, the authors extended this line of research and characterized the H2B-H2A histone variant, termed MV-varH2B-H2A, that is conserved across all Marseilleviridae. Their functional assay shows that MV-varH2B-H2A benefits the Melbournevirus infection against amoeba. This led them to characterize the cryo-EM structure of the nucleosome-like particle containing this variant. Intriguingly, the MV-varH2B-H2A nucleosome only wraps 90 bp DNA, even shorter than the canonical Melbournevirus nucleosome. In addition, they built model structures of the "columnar nucleosomes" that contain MV-varH2B-H2A and canonical Melbournevirus histones and performed MD simulation to assess whether Melbournevirus can maintain the columnar nucleosomes, which may contribute the packaging of the Melbournevirus genome in Capsids.

Overall, this manuscript provides new insights into the roles of histone variants in non-eukaryotes, significantly expanding our understanding of epigenetics. However, due to limitations of the paper, it remains unclear whether the MV-varH2B-H2A is an essential gene of Melbournevirus as stated by the authors. Additionally, the reported *in silico* model using columnar-telomeric nucleosomes as a template lacks experimental evidence to validate its occurrence and formation. Therefore, it is challenging to acknowledge the validity of this model being representative of the Melbournevirus-histone variant in viral genome organization. The following issues should be addressed before publication:

Major comments:

1). Essentiality of mel_149 (MV-varH2B-H2A) is unclear.

While the authors claimed that the MV-varH2B-H2A is essential for MV (e.g., line 128, the title of Figure 2), the competition experiment shown in Figure 2 does not provide enough evidence to support that idea. First, the method used in this experiment has to be described in the method section. Second, the loss of recombinant virus can happen by the 'bottleneck effect'. Indeed, sudden loss of the "recombinant" virus was observed in P2 to P3 in the complemented *A. castellanii* as well. To confirm if the loss of the recombinant virus was caused by the effect on viral fitness rather than random selection, the experiment has to be repeated multiple times. Third, the amount of the recombinant virus is much lower than the parental virus (Fig. 2A). Assuming from Fig 2B description and their previous paper (PMID: 34297924), authors might have selected the recombinant virus through the Nourseothricin-resistance gene (*nat1*) and therefore authors might be able to use the comparable amount of the recombinant and parental virus for this experiment. Fourth, to claim essentiality, the competition experiment described in Figure 2 may not be appropriate. Authors can simply test if the recombinant virus loses the ability to infect wildtype *A. castellanii* while maintaining the ability to infect complemented *A. castellanii*.

2). Structural and biochemical analyses.

The cryo-EM of the MV-varH2B-H2A containing nucleosome with 601 DNA suggested that this variant can form a nucleosome-like structure. However, in addition to histone variation, nucleosome structure may be affected by the DNA

sequence it harbors. Although 601 DNA is known to be the strongest positioning sequence that facilitates nucleosome formation, MV-varH2B-H2A containing nucleosome only wraps the 90 bp DNA. This may raise the question of whether MV-varH2B-H2A can even form nucleosomes on native Melbournevirus genomic DNA. The author should address this point by citing previous papers (e.g., PMID: 36370708) or conducting experiments.

The cryo-EM of the MV-varH2B-H2A nucleosome also suggested that the DNA wrapping of this nucleosome is loose compared to the canonical Melbournevirus nucleosome. Since DNA entry/exit structure can vary depending on crosslink level and/or the effect of the air-water interface, this observation should be supported by another experiment, such as a nuclease sensitivity assay. Although the authors performed the thermal stability assay (Fig E2C) and showed that the MV-varH2B-H2A nucleosome is more unstable than canonical Melbournevirus nucleosome, the thermal stabilities of the nucleosomes are not always consistent with DNA wrapping. For instance, human H2A.B nucleosome, which wraps shorter DNA (PMID: 15257289), has higher thermal stability than H2A nucleosome (PMID: 25220913).

3). MD Simulation.

The current manuscript does not provide enough data to validate their MD simulation result (e.g., The atomic models before, after, and during the simulation should be presented. The definition of "DNA truncated", "r.m.s.d", and "distance" has to be stated. They should provide at least one example that this simulation has the ability to predict the dissociation of "columnar" nucleosome stacking [For instance, if the DNA sequence of 7v96 is replaced with the 601 sequences from (TTAGGG)_n, will human nucleosome dissociate columnar nucleosome?]). It is also unclear whether their MD simulation, in which canonical and variant Melbournevirus nucleosomes can maintain "columnar" nucleosome stacking, accurately recaptures the biophysical feature of the MV-varH2B-H2A containing nucleosome. While the previous paper by the Henikoff group suggested that the nucleosome repeat length of the Melbournevirus is short (PMID: 36370708), it does not indicate the formation of the columnar nucleosome. As the author's group showed previously with Archaeal nucleosomes, adjacent short linker DNA nucleosomes can be arranged 90° out-of-plane with one another (PMID: 33650488). Without the biochemical evidence that supports the formation of the columnar nucleosome by Melbournevirus histones, showing the MD simulation using the columnar nucleosome as a starting model does not make sense.

As the manuscript reports variable findings without having this simulation, the authors should consider removing the MD simulation result from this manuscript. Although the authors may be able to add structural and biochemical analyses and MD simulations to confirm their observation (e.g., Showing the biological evidence to support the simulation result. Identifying the critical mechanism that causes the packing, simulating with the mutant, and performing a biochemical assay with the mutant), that may require several years of research.

Minor comments:

1. Fig 2A: 'P1+b' under P3 might be a typo.
2. Fig 2A: It may be helpful to explain the
3. Line 146 and Figure E2 could benefit by stating the exact eukaryotic nucleosome that is being used in the assay for the comparison. Elaborating why human nucleosomes, which only have ~30% sequence identity between Melbournevirus nucleosomes, can be the right control to be compared may also be helpful.
4. Fig 5: Although their cryo-EM data suggest the MV-varH2B-H2A wraps 90 bp DNA, significantly shorter than the Melbournevirus canonical nucleosome, the starting model of their MD simulation does not reflect their own finding. They should provide the reason for it,

Reviewer #2

(Remarks to the Author)

The authors identified Melbournevirus H2A-H2B histone variants, reconstituted a nucleosome, and analyzed it using thermal stability assay and cryo-EM analysis. They found that variant nucleosomes form only 90bp DNA-wrapped nucleosomes and missing some interactions. In the end, the authors performed the MD simulation to evaluate the inter-nucleosomal interaction, and variant di-nucleosomes are stable like the canonical di-nucleosome.

This is the first time a histone variant has been shown in the virus. In eukaryotes, histone variants have significant functional roles. Therefore, this finding is very important. However, I feel that some experiments and manuscript rewriting are required before accepting the manuscript to Nature Communications.

Here are my comments;

1. The cryo-EM structural data (9CVT) shows it forms a much shorter DNA wrapping size than the canonical virus nucleosome. However, the thermal assay shows similar stability. The authors need to describe the discrepancy or to prove by another experiment, such as MNase assay.
2. Line 121-line 123 "As was observed for the deletion of MV-H4-H3, the recombinant viruses were quickly lost and outcompeted by wild-type virus." Can the authors show the data? The authors may want to rephrase "recombinant" to "variant H2B-H2A deleted".
3. It is related to no 2, and it was not clear to me in Figure 2. I think the authors should consider remaking Figure 2 to describe the necessity of the variant H2B-H2A for Melbournevirus.
4. The authors used the MD simulation to suggest the di-nucleosome compaction. The data suggests that the variant di-nucleosome is similar to the canonical di-nucleosome. The authors used the telomeric di-nucleosome as a comparison since it has less linker size. Is the virus genomic sequence similar to the telomeric sequence in terms of GC contents? Does the author think whether the structure of virus nucleosome in virus genomic DNA is similar to the structures made by the 601 sequence? Is it possible to prove di-nucleosome interactions by some experiments?

The minor comments

1. Figure 5 labelling “MV-varNLP “ should be “MV-valNLP(9cvt)”.
2. The number of positively charged amino acids in variants is very limited, and the amino acid sequences in other species are quite similar. Could the authors provide more details in the manuscript?

Reviewer #3

(Remarks to the Author)

In the manuscript titled “Melbournevirus encodes a shorter H2B-H2A doublet histone variant that forms structurally distinct nucleosome structures”, Villalta et al. reported cryo-EM structure of a nucleosome-like particle reconstituted with viral H4-H3 and a newly identified variant H2B-H2A doublet in Melbournevirus. The authors observed several interesting features of the MV-var-NLP which are different from the canonical MV-NLP, including disulfate bond, binding only 90 base pairs of DNA. Moreover, the authors have shown that MV-varH2B-H2A (mel_149) is essential for the survival of *A. castellanii* in Figure 2. These findings have the potential to open new window to the regulatory mechanisms of viral genome structure and function. However, the authors stopped at describing the structural differences and did not go further to establish a causal relationship between these structural differences and functions, which makes the biological significance of these findings unsupported. Based on the above considerations, I think that this manuscript is not yet qualified for publication in a high-impact journal such as Nature communications.

Major points:

As the histones and telomeric DNA formed structurally different chromatin from those with 601 DNA, is it possible that MV-histones form different NLP when with melbournevirus DNA compare with Widom 601 DNA?

Minor points:

As mel_149 has no recognizable homology to histone H2B, and only ~ 40% identical to H2A, is it possible that it is only a gene with histone fold? Did your blast reveal other proteins with high similarity?

I think the readability of Figure 2 needs to be improved. Perhaps Figure 2B can be explained ahead of Figure 2A, then “a+b” and “b+c” can be indicated in Figure 2A. Can you indicate what is “complemented”? Does “Parental” refers to melbournevirus or amoebae in Figure 2A? The real DNA marker bands should be shown. One band size is missing.

Is “*Acanthamoeba castellanii*” at Line 121 the “amoebae” at line 124? Is “*A. castellanii*” in the legend of Figure 2 the same thing?

At line 123, figure indication is missing at “..... outcompeted by wild-type virus.”

Is there a sub-title at line 192-193?

Is the sentence “the lighter area around the mean curve correspond to the standard variation around the mean curve” at line 282-283 repeated at line 286-287?

In Figure 5, while “DNA truncated” is labeled for C and D, what should be labeled for A and B? While there is a di-nucleosome carton at figure B, what should be shown for A, C and D?

In the figure legend of A and B, the same color is used for MV-varNLP and MV-hybridNLP.

Version 1:

Reviewer comments:

Reviewer #1

(Remarks to the Author)

The authors addressed most of the concerns I (Reviewer #1) pointed out in the first round of the review. However, there is still a concern about MD simulation.

It is unclear if the MD simulation data can be used to conclude that “di-nucleosomal stacks can be formed” because it is still unclear if the MD simulation has the ability to predict the dissociation of columnar nucleosome stacking. Although the authors provided new data simulated with H4-H3 tailless nucleosomes and half of the replicates detected the increased “distance” (Extended Figure 7F and L), the “r.m.s.d.” didn’t change that much (Extended Figure 7E and K). Seeing the currently presented data, this simulation seems to cluster everything together and is not suitable for assessing whether di-nucleosomal stacks can be formed or not.

The authors should consider removing the MD simulation result from this manuscript or testing this MD simulation with a

more drastic negative control (e.g., putting a random protein on a nucleosome or mutating all surface residues to Asp).

Reviewer #2

(Remarks to the Author)

The authors have addressed the majority of my concerns. I do not have any further comments.

Reviewer #3

(Remarks to the Author)

The authors addressed all my concerns, now I support the publication of this manuscript.

In figure 2C, unit (Kb ?) is missing for the band size.

We appreciate the reviewer's time and effort in reviewing our manuscript. Their valuable comments and suggestions have helped us improve the clarity and quality of our work. Below, we provide detailed responses to each point and outline the corresponding revisions (shown in blue in the revised version of the manuscript). We hope these changes address your concerns and you now find our manuscript suitable for publication.

Main changes

1. We have revised figure 2 and have added more explanation to both the text and the figure legend.
2. We have added a new figure panel (extended figure 4) describing an MV-varNLP on a generic DNA unrelated to 601 DNA. This particle looks very similar than what is formed on 601 DNA.
3. We have added additional MD simulations to demonstrate that removal of the histone tails results in significant destabilization of the dinucleosome stacks.

Reviewer #1 (Remarks to the Author):

The nucleosome is the basic structural unit of the eucaryotic chromosomes. The author's group has been pioneering the structural biology of the nucleosomes and nucleosome-containing complexes. The authors and other groups have recently found that non-eucaryotes, such as Archaea and Marseillevirus, also have nucleosomes (PMID: 28798133, 34297924, 33927388). In the Marseillevirus, histones are encoded as doublets, linking H4 to H3 and H2B to H2A, and form nucleosomes that only wrap around ~120 bp DNA. In this manuscript, the authors extended this line of research and characterized the H2B-H2A histone variant, termed MV-varH2B-H2A, that is conserved across all Marseilleviridae. Their functional assay shows that MV-varH2B-H2A benefits the Melbournevirus infection against amoeba. This led them to characterize the cryo-EM structure of the nucleosome-like particle containing this variant. Intriguingly, the MV-varH2B-H2A nucleosome only wraps 90 bp DNA, even shorter than the canonical Melbournevirus nucleosome. In addition, they built model structures of the "columnar nucleosomes" that contain MV-varH2B-H2A and canonical Melbournevirus histones and performed MD simulation to assess whether Melbournevirus can maintain the columnar nucleosomes, which may contribute the packaging of the Melbournevirus genome in Capsids.

Overall, this manuscript provides new insights into the roles of histone variants in non-eukaryotes, significantly expanding our understanding of epigenetics. However, due to limitations of the paper, it remains unclear whether the MV-varH2B-H2A is an essential gene of Melbournevirus as stated by the authors. Additionally, the reported in silico model using columnar-telomeric nucleosomes as a template lacks experimental evidence to validate its occurrence and formation. Therefore, it is challenging to acknowledge the validity of this model being representative of the Melbournevirus-histone variant in viral genome organization. The following issues should be addressed before publication:

Major comments:

1). Essentiality of mel_149 (MV-varH2B-H2A) is unclear.

While the authors claimed that the MV-varH2B-H2A is essential for MV (e.g., line 128, the title of Figure 2), the competition experiment shown in Figure 2 does not provide enough evidence to support that idea. First, the method used in this experiment has to be described in the method section. Second, the loss of recombinant virus can happen by the 'bottleneck effect'. Indeed, sudden loss of the "recombinant" virus was observed in P2 to P3 in the complemented *A. castellanii* as well. To confirm if the loss of the recombinant virus was caused by the effect on viral fitness rather than random selection, the experiment has to be repeated multiple times. Third, the amount of the recombinant virus is much lower than the parental virus (Fig. 2A). Assuming from Fig 2B description and their previous paper (PMID: 34297924), authors might have selected the recombinant virus through the Nourseothricin-resistance gene (*nat1*) and therefore authors might be able to use the comparable amount of the recombinant and parental virus for this experiment. Fourth, to claim essentiality, the competition

experiment described in Figure 2 may not be appropriate. Authors can simply test if the recombinant virus loses the ability to infect wildtype *A. castellanii* while maintaining the ability to infect complemented *A. castellanii*.

We were unable to obtain pure or clonal recombinant viruses of MV-varH2B-H2A, which is why we used a mixed population. This mixed population was enriched through selection with nourseothricin (for 4 passages), but further enrichment could not be achieved, either by increasing the number of passages with the selection or by cloning via limited dilution. Notably, this mixed progeny was only obtained in trans-complementing cells, which strongly suggests that the gene is crucial for the Melbournevirus infectious cycle (to answer point 3).

To address point 1, we provided a more detailed description. To further characterize the phenotype, we removed the selection and performed additional infections with the enriched mixed population. The recombinant viruses were eventually lost in both trans-complemented and non-trans-complemented lines, but wild-type viruses out-competed the recombinant viruses more rapidly in non-trans-complemented cells (observed in two independent experiments), indicating that the loss was not due to a bottleneck effect (specifically addressing point 2).

Regarding point 4, we agree with the reviewer that while our results strongly suggest that MV-varH2B-H2A is likely essential, we cannot fully confirm this. Therefore, we have revised the manuscript to describe the gene as "likely-essential." While we were eager to further investigate the role of MV-varH2B-H2A, the lack of inducible systems for manipulating gene deletion or expression presents a major technical hurdle for studying essential genes. We hope to overcome this challenge in the near future.

In response to this and reviewer 2 comments, we revised the text on page 4 (shown in blue in the revised manuscript), and revised Figure 2 and the figure legend.

2). Structural and biochemical analyses.

The cryo-EM of the MV-varH2B-H2A containing nucleosome with 601 DNA suggested that this variant can form a nucleosome-like structure. However, in addition to histone variation, nucleosome structure may be affected by the DNA sequence it harbors. Although 601 DNA is known to be the strongest positioning sequence that facilitates nucleosome formation, MV-varH2B-H2A containing nucleosome only wraps the 90 bp DNA. This may raise the question of whether MV-varH2B-H2A can even form nucleosomes on native Melbournevirus genomic DNA. The author should address this point by citing previous papers (e.g., PMID: 36370708) or conducting experiments.

While the wrapping of the DNA around the histone core is NOT affected by DNA sequence, we agree with the reviewers that the unique properties of the 601 sequence might give rise to misleading results in terms of stability, and how DNA much is wrapped. The same concern was also raised by reviewer 3. We now provide evidence for a MV-varH2B-H2A-containing nucleosome on a generic 150 bp DNA sequence that mimics the GC content of viral DNA (new Extended Figure 4; see also response to reviewer 3), and that has none of the unique properties of the 601 sequence. As such, we believe that the MV-varNLP structure is formed independently of DNA sequence.

The cryo-EM of the MV-varH2B-H2A nucleosome also suggested that the DNA wrapping of this nucleosome is loose compared to the canonical Melbournevirus nucleosome. Since DNA entry/exit structure can vary depending on crosslink level and/or the effect of the air-water interface, this observation should be supported by another experiment, such as a nuclease sensitivity assay. Although the authors performed the thermal stability assay (Fig E2C) and showed that the MV-varH2B-H2A nucleosome is more unstable than canonical Melbournevirus nucleosome, the thermal stabilities of the nucleosomes are not always consistent with DNA wrapping. For instance, human H2A.B nucleosome, which wraps shorter DNA (PMID: 15257289), has higher thermal stability than H2A nucleosome (PMID: 25220913).

Our wording was somewhat misleading. We indeed see very weak density that continues on the ‘canonical’ path along the MV-varH2B-H2A surface, but the wrapping is likely much more dynamic, and more ‘breathing’ is observed. This is also something that is observed with many eukaryotic H2A variant structures. We had made this point in the original version. As such, we do not imply that DNA wrapping and thermal stability are necessarily related. We deleted this statement (line 357). Indeed, we were surprised that these particles weren’t more destabilized, and as we point out in the manuscript (line 365), we think this is due to the disulfide ‘staple’ that links the two MV-varH2B-H2A dimers together.

Nuclease sensitivity experiments on reconstituted MV-varNLP have already been performed by Bryson et al (PMID: 36370708; Figure 2C), demonstrating reduced resistance of MV-varNLP to MNase digestion, consistent with our results. We have now referred to this data on page 7.

3). MD Simulation.

The current manuscript does not provide enough data to validate their MD simulation result (e.g., The atomic models before, after, and during the simulation should be presented. The definition of “DNA truncated”, “r.m.s.d”, and “distance” has to be stated.

We are depositing movies of the entire simulation and the starting models on figshare.

We modified the figure legend for Figure 5 to define the terms pointed out above, to make clear what was done to the di-nucleosomes before the simulation.

We modified page 7 to define r.m.s.d.

They should provide at least one example that this simulation has the ability to predict the dissociation of “columnar” nucleosome stacking [For instance, if the DNA sequence of 7v96 is replaced with the 601 sequences from (TTAGGG)_n, will human nucleosome dissociate columnar nucleosome?]).

We have performed a simulation for the viral nucleosomes where we removed the H2B N-terminal tail that in our simulation bridges two nucleosomes. In both cases, the columnar nucleosome stacks are destabilized (according to the increase in the distance between the two nucleosomes during the simulation). This suggests that it is unlikely that the DNA sequence (or properties of the 601 DNA) artificially holds the stacks together, and demonstrates the ability of MD simulations to observe the dissociation of the di-nucleosome stacks. This data has been added to Extended Figure 7. We have no reason to believe that exchanging the DNA sequence would change the trajectory of the simulation.

It is also unclear whether their MD simulation, in which canonical and variant Melbournevirus nucleosomes can maintain “columnar” nucleosome stacking, accurately recaptures the biophysical feature of the MV-varH2B-H2A containing nucleosome. While the previous paper by the Henikoff group suggested that the nucleosome repeat length of the Melbournevirus is short (PMID: 36370708), it does not indicate the formation of the columnar nucleosome. As the author’s group showed previously with Archaeal nucleosomes, adjacent short linker DNA nucleosomes can be arranged 90° out-of-plane with one another (PMID: 33650488). Without the biochemical evidence that supports the formation of the columnar nucleosome by Melbournevirus histones, showing the MD simulation using the columnar nucleosome as a starting model does not make sense.

We agree with this reviewer that the columnar nucleosome structure is only one of several possible models, and it is not meant to represent the actual features of the nucleosome stack. We focused on this model since it is the model already proposed by Henikoff group in (PMID: 36370708), although their experiments were not designed to demonstrate the molecular details of these stacks. Nevertheless, the published structure of telomeric dinucleosomes provided a reasonable working model. Our point was that both types of viral nucleosomes are compatible with this type of stacking, where histone surfaces come in close contact with each other. Had they

not been, we would not have been able to draw any conclusions, but our data show that indeed dinucleosomal stacks can be formed.

We modified the text on page 11 and 12 to make our reasoning clearer.

As the manuscript reports variable findings without having this simulation, the authors should consider removing the MD simulation result from this manuscript. Although the authors may be able to add structural and biochemical analyses and MD simulations to confirm their observation (e.g., Showing the biological evidence to support the simulation result. Identifying the critical mechanism that causes the packing, simulating with the mutant, and performing a biochemical assay with the mutant), that may require several years of research.

We agree that verifying the putative model of the dinucleosome (or, indeed columnar stacks) with 'real' structures would be of high interest, but in our opinion exceeds the scope of this paper. In the interim, we believe that our simulations are of value because they clearly show that both types of MV-NLPs are consistent with this type of packing, i.e. there is nothing in the structure that would preclude the close packing of nucleosomes in this manner. We believe this is a finding that can guide future research and we would therefore like to share this with the readers. We added a statement to qualify our findings on page 12.

Minor comments:

1. Fig 2A: 'P1+b' under P3 might be a typo.
2. Fig 2A: It may be helpful to explain the
3. Line 146 and Figure E2 could benefit by stating the exact eukaryotic nucleosome that is being used in the assay for the comparison. Elaborating why human nucleosomes, which only have ~30% sequence identity between Melbournevirus nucleosomes, can be the right control to be compared may also be helpful.

Points 1 is fixed, thank you. point 2 is truncated...

point 3: We added on line 170: "The human nucleosome was used as a control as a proxy for a stably folded 'canonical' nucleosome."

4. Fig 5: Although their cryo-EM data suggest the MV-varH2B-H2A wraps 90 bp DNA, significantly shorter than the Melbournevirus canonical nucleosome, the starting model of their MD simulation does not reflect their own finding. They should provide the reason for it,

We thank this reviewer for pointing out this discrepancy. Although we only see 90 bp stably bound on the structure, additional weak density is seen beyond the 90 bp following the canonical path (extended Figure 5). As such we do believe that in a chromatin context (especially in the context of closely packed nucleosomes, as demonstrated by Bryson et al.), more than 90 bp will be bound to the nucleosome, which justified the use of this starting model. We must emphasize that the main point was to observe whether there are any features on the histone surface that might preclude packing of viral nucleosomes, which is not the case. We have added a statement to this effect on line 394.

Reviewer #2 (Remarks to the Author):

The authors identified melbournevirus H2A-H2B histone variants, reconstituted a nucleosome, and analyzed it using thermal stability assay and cryo-EM analysis. They found that variant nucleosomes form only 90bp DNA-wrapped nucleosomes and missing some interactions. In the end, the authors performed the MD simulation to evaluate the inter-nucleosomal interaction, and variant di-nucleosomes are stable like the canonical di-nucleosome.

This is the first time a histone variant has been shown in the virus. In eukaryotes, histone variants have significant functional roles. Therefore, this finding is very important. However, I feel that some experiments and

manuscript rewriting are required before accepting the manuscript to Nature Communications.

Here are my comments;

1. The cryo-EM structural data (9CVT) shows it forms a much shorter DNA wrapping size than the canonical virus nucleosome. However, the thermal assay shows similar stability. The authors need to describe the discrepancy or to prove by another experiment, such as MNase assay.

This is a comment already addressed by reviewer 1 (partially answered) . “The thermal stabilities of the nucleosomes are not always consistent with DNA wrapping. For instance, human H2A.B nucleosome, which wraps shorter DNA (PMID: 15257289), has higher thermal stability than H2A nucleosome (PMID: 25220913)”. This reviewer can see our response to both reviewers addressed above.

2. Line 121-line 123 “As was observed for the deletion of MV-H4-H3, the recombinant viruses were quickly lost and outcompeted by wild-type virus.” Can the authors show the data? The authors may want to rephrase “recombinant” to “variant H2B-H2A deleted”.

The data showing the outcompeting of MV-H4-H3 deleted viruses by wild type viruses has been already published in our earlier paper (Liu et al 2021). We modified the text to add the reference to this experiment that we did not mention in the previous version.

We changed the term “recombinant” for “variant H2B-H2A deleted”

3. It is related to no 2, and it was not clear to me in Figure 2. I think the authors should consider remaking Figure 2 to describe the necessity of the variant H2B-H2A for Melbournevirus.

We redesigned figure 2 and further explained the experiment in the text and figure legend. We hope the new version is clearer.

4. The authors used the MD simulation to suggest the di-nucleosome compaction. The data suggests that the variant di-nucleosome is similar to the canonical di-nucleosome. The authors used the telomeric di-nucleosome as a comparison since it has less linker size. Is the virus genomic sequence similar to the telomeric sequence in terms of GC contents?

The telomeric sequence has a GC content of 50.2% while the overall GC content of MV genome is 44.7%. We have also added data showing that both viral nucleosomes can form on a generic DNA sequence with a GC content of 50 % (new extended Figure 4).

Does the author think whether the structure of virus nucleosome in virus genomic DNA is similar to the structures made by the 601 sequence?

It has been proposed by the Henikoff group (PMID: 36370708) that these nucleosomes might form such structures, and we agree that this is a good starting model. This is why we focused our simulations on this model, using this sequence. The same authors also showed that all viral DNA is covered by nucleosomes, and as such we don't believe that the overall structure of the viral nucleosome is affected by DNA sequence (see new Extended Figure 4), although their regional stability might be affected. The emerging picture is that the only effects of DNA sequence are how much of the terminal DNA is stably bound; the overall geometry and histone-DNA contacts are not affected by the DNA sequence. Indeed, no base-specific contacts are made by any histone, and the

positioning property of the 601 and similar sequences stems simply from sequence pattern resulting in favorable or unfavorable bending energy to conform to the shape of the histone octamer.

Is it possible to prove di-nucleosome interactions by some experiments?

It is indeed possible, but would require a large amount of work that, in our opinion, exceeds the scope of this manuscript.

The minor comments

1. Figure 5 labelling “MV-varNLP “should be “MV-valNLP(9cvt)”.

We agree with this reviewer’s point to keep consistent labelling. Upon further consideration, we now decided to more accurately label the structures as ‘human telomere’, ‘MV-NLP’ and ‘MV-varNLP’ since in our simulations we are not using the pdb files as they were deposited, but added tails to 7v96, and constructed *in-silico* dinucleosomes from pdb files 7n8n and 9cvt.

2. The number of positively charged amino acids in variants is very limited, and the amino acid sequences in other species are quite similar. Could the authors provide more details in the manuscript?

We are not sure we understand this reviewer’s question. We show that the variant is conserved in other viral strains, and that one of the differences between MV-H2B-H2A and MV-varH2b-H2A is the absence of the positively charged C terminal tail on varH2B-H2A.

We modified the line 297- to specify the charge of this tail to help emphasize the charge pointed out by this reviewer.

Reviewer #3 (Remarks to the Author):

In the manuscript titled “Melbournevirus encodes a shorter H2B-H2A doublet histone variant that forms structurally distinct nucleosome structures”, Villalta et al. reported cryo-EM structure of a nucleosome-like particle reconstituted with viral H4-H3 and a newly identified variant H2B-H2A doublet in Melbournevirus. The authors observed several interesting features of the MV-var-NLP which are different from the canonical MV-NLP, including disulfate bond, binding only 90 base pairs of DNA. Moreover, the authors have shown that MV-varH2B-H2A (mel_149) is essential for the survival of *A. castellanii* in Figure 2. These findings have the potential to open new window to the regulatory mechanisms of viral genome structure and function. However, the authors stopped at describing the structural differences and did not go further to establish a causal relationship between these structural differences and functions, which makes the biological significance of these findings unsupported. Based on the above considerations, I think that this manuscript is not yet qualified for publication in a high-impact journal such as Nature communications.

Causal relationship between structure and function is the holy grail of structural biology. Given the challenges of working on the melbournevirus system, the biological role of histones in giant viruses is not proven but most likely play a central role in genome packaging and packing in the virions. We believe that our finding that MV-varH2B-H2A indeed appears to be required for viral fitness, and cannot be complemented by the main MV-H2B-H2A, is a first step in a better understanding of the role of the different histones for this task.

Major points:

As the histones and telomeric DNA formed structurally different chromatin from those with 601 DNA, is it possible that MV-histones form different NLP when with melbournevirus DNA compare with Widom 601 DNA?

We share the concerns voiced by this reviewer and reviewer 1 about the idiosyncrasies of 601 DNA. As shown in our previous papers (Liu et al. 2021 and Toner et al., 2028), viral histones can also form nucleosomes with other

types of DNA. Widom 601 DNA has the advantage that it positions more precisely, leading to more uniform nucleosome populations on the grid. Thanks to both reviewers, we now show that we can form MV-varNLP nucleosomes with a 150 bp random DNA (GC-content of 50% which is close to the GC-content of 45% of melbournevirus genome). We collected a small data set on our Titan Krios and the 2-D class averages look remarkably similar to the particles formed with 601 DNA, however, the quality of this sample was not sufficient (compared to 601 DNA) for us to solve the structure of this sample. We believe that the 2-D class averages provide sufficient evidences that the 601 DNA sequence does not provide an artificial system but rather reflects the variant NLP structure on generic or viral DNA. This is now presented in the new extended figure 4, and the text on page 7 has been extended, as materials and methods to reflect the added data.

Minor points:

As mel_149 has no recognizable homology to histone H2B, and only ~ 40% identical to H2A, is it possible that it is only a gene with histone fold? Did your blast reveal other proteins with high similarity?

A blast on melbournevirus genome with H2B-H2A only retrieves two sequences (H2B-H2A and varH2B-H2A); when blasting the melbournevirus genome with varH2B-H2A, only varH2B-H2A is retrieved. The question of whether varH2B-H2A is indeed a histone is the central question answered by our paper, and confirms statements made in Bryson et al. It forms nucleosomes with the other histone doublet, and is bound to melbournevirus chromatin (PMID: 36370708). As such, it has all the hallmarks of a *bona fide* histone variant.

I think the readability of Figure 2 needs to be improved. Perhaps Figure 2B can be explained ahead of Figure 2A, then “a+b” and “b+c” can be indicated in Figure 2A. Can you indicate what is “complemented”? Does “Parental” refers to melbournevirus or amoebae in Figure 2A? The real DNA marker bands should be shown. One band size is missing.

We have incorporated all the changes in a revised and hopefully improved figure 2.

Is “Acanthamoeba castellanii” at Line 121 the “amoebae” at line 124? Is “A. castellanii” in the legend of Figure 2 the same thing?

We understand the confusion of this reviewer and we changed the text to refer to *Acanthamoeba castellanii* as *A. castellanii* consistently through the text after presenting the nomenclature.

At line 123, figure indication is missing at “..... outcompeted by wild-type virus.”
Thank you for pointing this out. The reference has been added

Is there a sub-title at line 192-193?

It is not a sub-title but a problem of formatting, which has now been fixed

Is the sentence “the lighter area around the mean curve correspond to the standard variation around the mean curve” at line 282-283 repeated at line 286-287?

Thank you for catching this, we deleted the duplicated sentence

In Figure 5, while “DNA truncated” is labeled for C and D, what should be labeled for A and B? While there is a di-nucleosome carton at figure B, what should be shown for A, C and D?

In the figure legend of A and B, the same color is used for MV-varNLP and MV-hybridNLP.

We apologize for the confusion with figure 5, we changed it accordingly. We changed the colors for MV-varNLP and MV-hybridNLP

We thank the reviewers for a quick turnaround. We believe that we were able to address the lingering concerns by performing a 'negative control' for our MD simulations, demonstrating that di-nucleosomal stacks indeed will dissociate if interactions are unfavorable (added simulation in Figure 5B and Extended Figure 7).

Reviewer #1 (Remarks to the Author):

The authors addressed most of the concerns I (Reviewer #1) pointed out in the first round of the review. However, there is still a concern about MD simulation.

It is unclear if the MD simulation data can be used to conclude that "di-nucleosomal stacks can be formed" because it is still unclear if the MD simulation has the ability to predict the dissociation of columnar nucleosome stacking. Although the authors provided new data simulated with H4-H3 tailless nucleosomes and half of the replicates detected the increased "distance" (Extended Figure 7F and L), the "r.m.s.d." didn't change that much (Extended Figure 7E and K). Seeing the currently presented data, this simulation seems to cluster everything together and is not suitable for assessing whether di-nucleosomal stacks can be formed or not.

The authors should consider removing the MD simulation result from this manuscript or testing this MD simulation with a more drastic negative control (e.g., putting a random protein on a nucleosome or mutating all surface residues to Asp).

We have taken the reviewer's suggestion and repeated MD simulations with more drastic negative controls, by either mutating residues that contact DNA and any amino acids in the neighboring nucleosome to Asp (one replicate). In a more conservative approach, we replaced select inter-nucleosome contact residues with Glutamate (H4-H3: amino acids 1-23; amino acids 94-131, and amino acids 157-160 to Glu; MV-varH2B-HH2A: amino acids 66-72 to Glu. We performed these simulations with intact DNA (shown in grey in Figure 5, and Supplementary figure 7) and with severed DNA. All of these disturbances resulted in major changes in inter-nucleosome distances upon simulation.

Reviewer #2 (Remarks to the Author):

The authors have addressed the majority of my concerns. I do not have any further comments.
Thank you!

Reviewer #3 (Remarks to the Author):

The authors addressed all my concerns, now I support the publication of this manuscript.

In figure 2C, unit (Kb ?) is missing for the band size.

Thank you! now fixed.

In the manuscript titled “Melbournevirus encodes a shorter H2B-H2A doublet histone variant that forms structurally distinct nucleosome structures”, Villalta *et al.* reported cryo-EM structure of a nucleosome-like particle reconstituted with viral H4-H3 and a newly identified variant H2B-H2A doublet in Melbournevirus. The authors observed several interesting features of the MV-var-NLP which are different from the canonical MV-NLP, including disulfate bond, binding only 90 base pairs of DNA. Moreover, the authors have shown that MV-varH2B-H2A (mel_149) is essential for the survival of *A. castellanii* in Figure 2. These findings have the potential to open new window to the regulatory mechanisms of viral genome structure and function. However, the authors stopped at describing the structural differences and did not go further to establish a causal relationship between these structural differences and functions, which makes the biological significance of these findings unsupported. Based on the above considerations, I think that this manuscript is not yet qualified for publication in a high-impact journal such as Nature communications.

Major points:

As the histones and telomeric DNA formed structurally different chromatin from those with 601 DNA, is it possible that MV-histones form different NLP when with melbournevirus DNA compare with Widom 601 DNA?

Minor points:

As mel_149 has no recognizable homology to histone H2B, and only ~ 40% identical to H2A, is it possible that it is only a gene with histone fold? Did your blast reveal other proteins with high similarity?

I think the readability of Figure 2 needs to be improved. Perhaps Figure 2B can be explained ahead of Figure 2A, then “a+b” and “b+c” can be indicated in Figure 2A. Can you indicate what is “complemented”? Does “Parental” refers to melbournevirus or amoebae in Figure 2A? The real DNA marker bands should be shown. One band size is missing.

Is “*Acanthamoeba castellanii*” at Line 121 the “amoebae” at line 124? Is “*A. castellanii*” in the legend of Figure 2 the same thing?

At line 123, figure indication is missing at “..... outcompeted by wild-type virus.”

Is there a sub-title at line 192-193?

Is the sentence “the lighter area around the mean curve correspond to the standard variation around the mean curve” at line 282-283 repeated at line 286-287?

In Figure 5, while “DNA truncated” is labeled for C and D, what should be labeled for A and B? While there is a di-nucleosome carton at figure B, what should be shown for A, C and D?

In the figure legend of A and B, the same color is used for MV-varNLP and MV-hybridNLP.